# Octogenarians Are the New Sexagenarians: Cruciate-Retaining Total Knee Arthroplasty Is Not Inferior to Posterior-Stabilized Arthroplasty in Octogenarian Patients

**DOI:** 10.3390/jcm11133795

**Published:** 2022-06-30

**Authors:** Riccardo D’Ambrosi, Prem Haridas Menon, Abhijeet Salunke, Ilaria Mariani, Giovanni Palminteri, Giuseppe Basile, Nicola Ursino, Laura Mangiavini, Michael Hantes

**Affiliations:** 1IRCCS Istituto Ortopedico Galeazzi, 20161 Milan, Italy; palminteri.giovanni92@gmail.com (G.P.); basiletraumaforense@gmail.com (G.B.); nicolaursino@libero.it (N.U.); laura.mangiavini@unimi.it (L.M.); 2Dipartimento di Scienze Biomediche per la Salute, Università Degli Studi di Milano, 20133 Milan, Italy; 3Division of Orthopedics and Traumatology, Government Medical College Trivandrum, Kerala University of Health Sciences, Thiruvananthapuram 695011, India; phmenon777@gmail.com; 4Gujarat Cancer Research Institute, Ahemadabad 380016, India; drabhijeetsalunke@gmail.com; 5Institute for Maternal and Child Health IRCCS “Burlo Garofolo”, 34137 Trieste, Italy; ilaria.mariani1618@gmail.com; 6Department of Orthopaedic Surgery, Faculty of Medicine, University Hospital of Larissa, University of Thessalia, 41110 Larissa, Greece; hantesmi@otenet.gr

**Keywords:** octogenarian, knee arthroplasty, knee osteoarthritis, posterior-stabilized, cruciate-retaining, survivorship

## Abstract

Purpose: The primary goal of this study was to compare survivorship and functional results in individuals aged 80 and over who underwent total knee arthroplasty (TKA) with cruciate-retaining (CR) or posterior-stabilized (PS) implants. Methods: We prospectively analyzed the clinical records of two consecutive cohorts for a total of 96 implants in patients aged 80 years or over. The first cohort consisted of 59 consecutive cemented PS cases, while the second cohort comprised 37 consecutive cemented CR cases. The decision to either perform a PS or CR arthroplasty was taken based on preoperative magnetic resonance imaging and intraoperative findings. The clinical evaluation entailed evaluating each patient’s visual analogue scale for pain (VAS), range of motion (flexion and extension), Knee Society Score (KSS), and Oxford Knee Score (OKS). Each patient was clinically evaluated the day before surgery (T_0_) and at two consecutive follow-ups at least 1 (T_1_) and 2 (T_2_) years after surgery. Implant survival was calculated using the Kaplan–Meier method. Results: Both groups showed statistically significant improvements at each follow-up compared with the preoperative values (*p* < 0.05). The CR group showed a higher flexion degree at T_1_ than the PS group (116.14 ± 5.57° versus 113.16 ± 7.66°; *p* = 0.048). No differences were found between the two groups regarding survival rate (chi-squared test *p*-value = 0.789). Three failures were noted in the CR group, while there were four in the PS group. Conclusions: This prospective clinical study demonstrates that CR and PS TKA had similar clinical outcomes in octogenarians with regard to knee function, postoperative knee pain, and other complications. Prosthesis survivorship for CR and PS TKA were both satisfactory, and in selected octogenarian patients, CR TKA should always be considered because of the reduced surgical time.

## 1. Introduction

Osteoarthritis (OA) is the world’s most common joint ailment and one of the leading causes of pain and impairment in the aged. Age appears to be the single largest risk factor for the development of primary OA in vulnerable joints, according to accumulating research [1].

People live longer across the world. Between 2020 and 2050, the number of people aged 80 and over is predicted to treble, reaching 426 million. Thus, managing the huge burden of osteoarthritis in this alarmingly aging population can prove to be a real challenge for healthcare professionals across the globe [2].

In the end, it is the substantial burden of pain that patients with OA bear that is of paramount concern. This burden can be fundamentally classified into pain and functional impairment, which when combined have a significant negative impact on an individual’s quality of life (QOL) [3].

Osteoarthritis predominantly affects the weight-bearing knee joints, and total knee arthroplasty (TKA) remains the standard-of-care treatment for advanced arthritis in patients in whom conservative measures have failed. Conventional primary TKA uses two designs: one in which the posterior cruciate ligament (PCL) is preserved and the other in which the PCL is sacrificed. The former is termed cruciate-retaining (CR) prosthesis, and the latter is known as a posterior-stabilized (PS) design. In PS designs, the PCL is excised, and the implant design used incorporates an intercondylar tibial prominence that can act as a substitute for serving the function of the PCL. In flexion, this part articulates with the femur, leading to a femoral roll-back. The possible advantage of the above mechanism is that it provides longer lever arm in the quadriceps mechanism, thereby providing a slightly greater range of motion. Various studies have supported the slight increase in the range of motion at the terminal flexion. Additionally, the femoral roll-back mechanism reduces shearing stress on the polyethylene liner. Cruciate-retaining implant designs, on the other hand, have a clear advantage in terms of knee stability and proprioception. In light of the retained PCL, studies on the cruciate-retaining group have suggested that this design has near-normal knee kinematics [4,5,6].

There is a paucity of reasearch on the long-term functional outcomes comparing both designs. However, a number of studies have compared the short- and medium-term functional results, and they have not demonstrated any significant difference in early functional outcomes. These studies have focused on patients aged 60–80 years old, with a small fraction of patients aged 80 and over. According to Swedish Knee Arthroplasty Register, the incidence of knee arthroplasty in patients aged 75 to 84 years old was 665.2/100,000 inhabitants for women and 546.4/100,000 for men [7].

There are few studies comparing functional outcomes in the exclusive 80+ age group, which is growing swiftly. As a result, we believe it is critical to examine the results in the 80+ age range [1,8,9]

The primary goal of this study was to compare survivorship and functional results in individuals aged 80 and over who underwent TKA surgery with CR or PS implants.

## 2. Materials and Methods

All the procedures in this study involving human participants followed the ethical standards of the institutional and/or national research committee as well as the 1964 Helsinki Declaration and its later amendments or comparable ethical standards. The study was conducted following the STROBE guidelines [10]. Informed consent was obtained from all the participants included in the study. Furthermore, the appropriate ethical approval was obtained from the local ethics committee. We prospectively analyzed the clinical records of two consecutive cohorts including a total of 96 implants in patients aged 80 years or over. The first cohort consisted of 59 consecutive cemented posterior-stabilized TKA, while the second cohort comprised 37 consecutive cemented cruciate-retaining TKA.

The inclusion criteria were as follows: 24-month minimum follow-up; TKA performed by a single surgeon; severe knee pain and disability requiring primary unilateral TKA based on physical exam and medical and radiographic history; ability to cooperate in the required postoperative therapy; ability to complete scheduled follow-up evaluations as described in the informed consent form.

The exclusion criteria were as follows: follow-up < 24 months; multiple comorbidities and neurological illnesses (e.g., diabetic neuropathy, multiple sclerosis, and lateral amyotrophic sclerosis); revision surgery; previous surgery to the affected knee (except arthroscopy for meniscectomy or anterior cruciate ligament (ACL) reconstruction).

The indication for TKA was either primary or secondary grade IV osteoarthritis of the knee. All preoperative radiographs were assessed using the Kellgren and Lawrence classification system [11].

Each patients underwent plain radiographies and magnetic resonance imaging (MRI) preoperatively. Patients were allocated to receive PS or CR implants. The allocation followed the anatomopathological features of PCL at imaging, clinical, and intraoperative findings. If the PCL was intact and effective, a CR TKA was implanted, while in case of the lack of or a deficient PCL, a PS TKA was implanted. A standard medial para-patellar approach was used in all patients. No tourniquet was used. All patients received a cemented Vanguard Knee System (Zimmer-Biomet, Warsaw, IN, USA) following the manufacturer’s instructions. Both tibial and femoral components were cemented using Refobacin^®^ Bone Cement R (Zimmer Biomet, Warsaw, IN, USA) [12].

All arthroplasties were performed with a medial para-patellar approach without the use of a tourniquet, and the cementation of both tibia and femur was performed in all patients.

### 2.1. Rehabilitation Protocol

Both patient groups followed the same rehabilitation protocol, which involved passive mobilization for both types of surgery; from day one, an active progressive mobilization of the joint and assisted walking with two crutches were started. Gradually and according to each patient, it was, therefore, recommended to increase the load during walking, continuing with isometric muscle-toning exercises until the total abandonment of walking aids.

### 2.2. Clinical Evaluation

All clinical assessments were performed by two independent clinicians who were not involved in the index surgery. The clinical evaluation entailed assessing each patient’s visual analogue scale for pain (VAS), range of motion (flexion and extension), Knee Society Score (KSS), and Oxford Knee Score (OKS). Each patient was clinically evaluated the day before surgery (T_0_) and at two consecutive follow-ups at least 1 (T_1_) and 2 (T_2_) years after surgery [13,14,15].

### 2.3. Survivorship

Implant survival was calculated using the Kaplan–Meier method. Revision was defined as the failure of the implant (infection, periprosthetic fracture) or death of the patient [16].

### 2.4. Statistical Analysis

An estimated sample of 74 subjects, 37 for each group, was required to compare the VAS values between CR and PS groups with a two-sided *t*-test, assuming a mean difference of 1, a standard deviation of 1.5 for both groups, 5% alpha, and 80% power. Given the same parameters, this sample also had 99% power to detect pre–post differences using a paired *t*-test, assuming a mean difference of 1, a standard deviation of 1.5, a correlation of 0.25 between measurements, and 5% alpha. Additional subjects were recruited to ensure statistical significance in case of adverse events.

### 2.5. Data Analysis

Summary statistics are presented as means and standard deviations (SDs) or absolute frequencies and percentages. After having tested the distribution of continuous variables, a *t*-test or, for categorical variables, a chi-squared test was performed to assess preoperative and clinical differences between CR and PS groups.

To examine the score differences between groups, first, a *t*-test was used to evaluate inter-group differences at each follow-up. Second, to assess differences based on time intervals in each group for each score, a mixed model analysis was performed, since it allowed us to consider correlations among repeated measures and to test the covariance structure. Compound symmetry, autoregressive models with both homogeneous and heterogeneous variances, and unstructured covariance structures were tested, and the unstructured covariance was evaluated as the best covariance structure using the likelihood ratio test and Akaike information criterion. Bonferroni adjustment was applied for multiple comparisons. Survival curves were estimated to assess differences in the failure rate between CR and PS groups. A Cox regression model was also performed using failure as the independent variable and group as a covariate. Lastly, according to the variable distribution, Pearson or Spearman correlations among the scores and the demographic and clinical variables, i.e., age, sex, and side, were estimated. All tests were two-sided, and *p*-values less than 0.05 were considered statistically significant. Statistical analyses were conducted in R version 4.1.1 (R Foundation for Statistical Computing, Vienna, Austria).

## 3. Results

### 3.1. Demographic Data

The two groups resulted homogeneous regarding all demographic and preoperative values (*p* > 0.05). The only statistically significant difference pertained to surgical time, which was significantly lower in the CR group (48.70 ± 7.83 min versus 56.19 ± 11.43 min; *p* = 0.001). All data are reported in Table 1.

### 3.2. Clinical Results

Both groups exhibited statistically significant improvements at each follow-up compared with the preoperative values (*p* < 0.05). The CR group showed a higher flexion degree at T_1_ than the PS group (116.14 ± 5.57° versus 113.16 ± 7.66°; *p* = 0.048). Detailed results are reported in Table 2.

### 3.3. Survival Analysis

No differences were found between the two groups regarding survival rate (chi-squared test *p*-value = 0.789). Three failures were noted in the CR group, while there were four in the PS group. Details on failures and risk survival are reported in Table 3 and Table 4 and Figure 1.

## 4. Discussion

There has been a large number of past studies that have compared the posterior-stabilized design with the cruciate-retaining design in patients undergoing total knee replacement surgery. However, these studies have focused on patients aged 75 years or under, with a very small fraction of patients aged 80 and over [4,5,6,8,9,17]. According to current data, the number of people aged 80 and over is expected to triple in the next two–three decades, making assessing the outcomes of TKA in this age group critical [2].

Many studies have demonstrated that PS designs provide a greater range of motion [4,9,18]. PS knees have a femoral roll-back mechanism that decreases shearing forces on the polyethylene liner, and this has helped to popularize the design [19,20]. However, studies on the CR group have suggested increased knee stability, proprioception, and near-normal kinematics in view of the retained PCL [19,20].

The decision to either retain or excise the PCL largely depends on the functional status of the PCL implant design and the discretion of the surgeon. Both of these designs are widely employed in current practice. The option to either excise or preserve the PCL in knees with a functional PCL is still debatable, and it comes down to the surgeon’s preference and experience [21].

A large number of studies have looked into the activities of everyday living and social life in relation to implant design. Rising from a chair, ascending and descending stairs, and walking on flat ground were among the variables analyzed in those studies, all of which we believe are critical in the 80+ age range. All of the above-mentioned characteristics showed considerable improvement in those studies. The CR and PS designs, on the other hand, did not show any significant variations [4,6].

With regard to which design is superior—the cruciate-retaining or posterior-stabilized design—there are not many data on the 80-year age group. The Cochrane Database of Systematic Reviews by Verra included 17 randomized and quasi-randomized controlled trials with 1810 patients and 2206 knees for analysis [22]. This was an extensive study wherein 15 outcome/subgroup titles were analyzed in depth. Only 4 of the 17 above-mentioned studies included patients in the age group of 80 and over. With respect to both designs, the study could not find any clinically significant differences with respect to pain, range of motion, or clinical and radiological outcomes. The study found that the posterior-stabilized design group had a greater range of motion and better knee functional scores that were statistically significant compared with the cruciate-retaining group, even though the authors stated that the above entities were not clinically relevant in view of the high variability in the data [22].

Demographic data such as age, sex, side, and surgical time were included in our study. Both the groups appeared homogenous with regard to demographic variables such as age, sex, and side. Concerning the surgical time, it was significantly lower in the CR group (48.70 min versus 56.19), which is well supported by the literature.

Preoperative scores were assessed in terms of the VAS score, flexion/extension of knee joint, and two outcome scores, the KSS and Oxford Knee Society scores. The two groups were homogeneous regarding all preoperative values.

The clinical evaluation of our patients included the visual analogue scale for pain (VAS), range of motion (flexion and extension), KSS, and OKS. The CR group showed a higher flexion degree than the PS group (at the one-year follow-up) even though the improvements in flexion movement became marginal at the two-year follow-up. This is a striking finding that we did not come across in any of the major studies highlighted below.

In a multi-centric prospective randomized clinical trial, Harato et al. performed a mid-term comparison of CR vs. PS knees in a cohort of 222 patients from seven major surgical centers around the world. At the 5-year follow-up, the CR group had a flexion of 113.7°, while the PS group had a flexion of 117.0°, with a statistically significant difference. This study, with the longest follow-up and comparable cohorts with respect to age, is in stark contrast with our finding that the CR group had a higher flexion at the short-term follow-up. The author highlighted the postoperative stiffness of knees; in total, 7 out of 99 CR knees and 1 out of 93 PS knee were found to be stiff, and this was attributed to the difference in kinematics in the two designs involved as the possible explanation [23].

Tanzer et al., in their study, found no differences in the range of motion between the CR and PS groups preoperatively or at any time postoperatively. However, in their study, the CR group had a mean age of 68 years, and the PS group had a mean age of 66 years, making it difficult to compare it with our study, where both the CR and PS groups had only patients in the over-80-year age group (35). In contrast to the above study, Maruyama et al. identified significant improvements in knee flexion after surgery only in PS knees, and the mean age of the study group was around 74.3 years (36). Wang et al. in their prospective clinical trial on 267 knees with an average participant age of 55 years found that PS knees had a higher flexion than CR knees at an average follow-up of 42 months [24].

All the above studies had patients in the age group 44–89 years, and they categorically stated that PS knees demonstrated a higher postoperative knee flexion than CR knees at short- and medium-term follow-ups. Only patients aged 80 years and over were included in our study, and our short-term results very clearly indicate a higher postoperative knee flexion in the CR group.

The postoperative gain in extension was the same with regard to CR and PS knees, which holds true with regard to the patients in the other studies here highlighted.

Previous studies have suggested that surgeons should prefer the PS design for TKA, especially to achieve a higher postoperative range of motion in patients with high functional demands, but in our cohort, all patients were octogenarians, so they were supposed to have a sedentary lifestyle [25].

Due to the lower physiological activity and lowered life expectancy, fears of revision surgery (and associated risks and complications) are minimized. The morbidity and complications associated with CR TKA, such as surgical time, and results in the form of pain relief and simpler rehabilitation provide high satisfaction levels [26]. The return to activities of daily living, which is perhaps most crucial in this age group of patients, is very rapid. Further, they pose a very low demand on the prosthetic knee, with the possibility of outliving the prosthetic knee.

Most authors have found that there is an increase in the risk of complications in patients aged over 80 years undergoing TKA. Kuperman [27], in his systematic review, vividly looked at the complications in this age group.

This article categorically states an increase in mortality in terms of myocardial infarction, pulmonary embolism and deep-vein thrombosis. A few papers have quoted a statistically significant trend of mortality [28,29], but a few studies have contradicted the above trend, considering it as being non-significant [30,31]. In regard to the comparison of the complications/survival rates of the CR and PS designs, there is a lack of data. In our study, no differences were found between the CR and PS groups regarding the survival rate, as suggested by the chi-squared test; furthermore, no differences were identified in the rate of complications between the two groups.

Our study had a few limitations. Only age-matched cohorts were considered in this study, and other variables were not considered. We only considered a short-term follow-up of two years, which we believe is a serious limitation. Anteroposterior and mediolateral laxity measurements were not considered in our study. A large majority of the studies we reviewed only had elderly populations aged 75 years and under.

## 5. Conclusions

This prospective clinical study demonstrates that CR and PS have similar clinical outcomes in octogenarians with regard to knee function, postoperative knee pain, and the other complications. Prosthesis survivorship for CR and PS are both satisfactory, and in selected octogenarian patients, CR should always be considered, taking into the reduced surgical time as well.

## Figures and Tables

**Figure 1 jcm-11-03795-f001:**
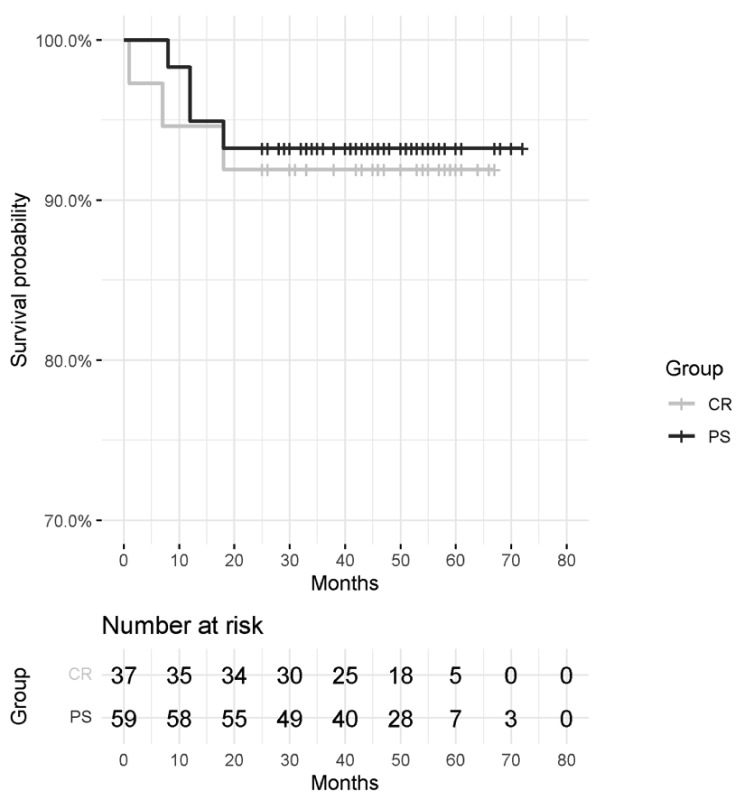
Kaplan-Meier curves and survival probabilities for each group.

**Table 1 jcm-11-03795-t001:** Pre- and intraoperative comparisons between groups.

	Group		
	CR *n* = 37 Mean ± SD	PS *n* = 59 Mean ± SD	*p*-Value
Age	82.30 ± 2.17	81.51 ± 1.83	0.063
Sex, *n* (%)			
F	27 (73.0)	43 (72.9)	1.000
M	10 (27.0)	16 (27.1)	
Side, *n* (%)			
Right	21 (56.8)	34 (57.6)	1.000
Left	16 (43.2)	25 (42.4)	
Surgical time	48.70 ± 7.83	56.19 ± 11.43	0.001 *
**Preoperative scores**			
VAS	7.46 ± 1.39	7.39 ± 1.26	0.800
Flexion	90.35 ± 8.20	90.36 ± 9.39	0.998
Extension	4.86 ± 3.22	4.93 ± 2.86	0.853
KSS	36.76 ± 9.30	36.75 ± 8.79	0.995
OKS	20.59 ± 4.13	20.80 ± 3.60	0.692
**Follow-up (months)**			
T_1_	13.00 ± 0.97	12.98 ± 1.09	0.743
T_2_	47.46 ± 12.26	47.35 ± 12.52	0.967

* statistically significant value *p* < 0.05. CR = cruciate-retaining; PS = posterior-stabilized; VAS = visual analogue scale for pain; KSS = Knee Society Score; OKS = Oxford Knee Score; T_1_ = minimum 1-year follow-up; T_2_ = minimum 2-year follow-up.

**Table 2 jcm-11-03795-t002:** Clinical comparisons between groups at each postoperative follow-up.

	CR Group	PS Group	Group Comparison	Time Comparison Adjusted *p*-Value			
	Mean ± SD	Mean ± SD	*p*-Value	CR		PS	
**VAS**							
T_0_	7.46 ± 1.39 (*n* = 37)	7.39 ± 1.26 (*n* = 59)	0.800	T_0_	T_1_	T_0_	T_1_
T_1_	1.66 ± 1.39 (*n* = 35)	2.05 ± 1.34 (*n* = 57)	0.179	<0.001 *	-	<0.001 *	-
T_2_	1.56 ± 1.33 (*n* = 34)	1.53 ± 1.02 (*n* = 55)	0.900	<0.001 *	>0.99	<0.001 *	0.002 *
**Flexion (°)**							
T_0_	90.35 ± 8.20 (*n* = 37)	90.36 ± 9.39 (*n* = 59)	0.998	T_0_	T_1_	T_0_	T_1_
T_1_	116.14 ± 5.57 (*n* = 35)	113.16 ± 7.66 (*n* = 57)	0.048*	<0.001 *	-	<0.001 *	-
T_2_	120.00 ± 3.69 (*n* = 34)	118.91 ± 4.27 (*n* = 55)	0.221	<0.001 *	0.007 *	<0.001 *	<0.001 *
**Extension (°)**							
T_0_	4.86 ± 3.22 (*n* = 37)	4.93 ± 2.86 (*n* = 59)	0.853	T_0_	T_1_	T_0_	T_1_
T_1_	1.14 ± 2.45 (*n* = 35)	2.11 ± 2.98 (*n* = 57)	0.093	<0.001 *	-	<0.001 *	-
T_2_	1.18 ± 2.48 (*n* = 34)	1.18 ± 2.35 (*n* = 55)	0.915	<0.001 *	>0.99	<0.001 *	0.014 *
**KSS**							
T_1_	36.76 ± 9.30 (*n* = 37)	36.75 ± 8.79 (*n* = 59)	0.995	T_0_	T_1_	T_0_	T_1_
T_2_	91.00 ± 5.53 (*n* = 35)	89.39 ± 7.08 (*n* = 57)	0.253	<0.001 *	-	<0.001 *	-
T_3_	91.62 ± 5.03 (*n* = 34)	91.09 ± 5.50 (*n* = 55)	0.652	<0.001 *	0.316	<0.001 *	0.002 *
**OKS**							
T_1_	20.59 ± 4.13 (*n* = 37)	20.80 ± 3.60 (*n* = 59)	0.692	T_0_	T_1_	T_0_	T_1_
T_2_	43.86 ± 1.77 (*n* = 35)	43.72 ± 2.03 (*n* = 57)	0.588	<0.001 *	-	<0.001 *	-
T_3_	44.09 ± 1.85 (*n* = 34)	43.98 ± 2.26 (*n* = 55)	0.772	<0.001 *	>0.99	<0.001 *	>0.99

* statistically significant value *p* < 0.05. CR = cruciate-retaining; PS = posterior-stabilized; VAS = visual analogue scale for pain; KSS = Knee Society Score; OKS = Oxford Knee Score; T_0_ = follow-up the day before surgery; T_1_ = minimum 1-year follow-up; T_2_ = minimum 2-year follow-up; T_3_ = minimum 3-year follow-up.

**Table 3 jcm-11-03795-t003:** Analysis of failures and complications for both groups during the study period.

	Group	
	CR *n* = 37	PS *n* = 59
Failures at T_1_		
Death	1 (7th month)	1 (8th month)
Infection	1 (1st month)	0
Periprosthetic Fracture	0	1 (12th month)
Failures at T_2_		
Death	1 (18th month)	2 (18th and 28th month)
Total number of failures	3	4

CR = cruciate-retaining; PS = posterior-stabilized.

**Table 4 jcm-11-03795-t004:** Analysis of survival in each study group.

Month	Number of Patients at Risk	Number of Failures	Survival Probability % (SE)
**CR group**			
1	37	1	97.3 (2.67)
7	36	1	94.6 (3.72)
18	35	1	91.9 (4.49)
**PS group**			
8	59	1	98.3 (1.68)
12	58	2	94.9 (2.86)
18	56	1	93.2 (3.27)
Difference in survival time: chi-squared test *p*-value = 0.789			

CR = cruciate-retaining; PS = posterior-stabilized.

## Data Availability

Raw data have been submitted as Appendix A to the Journal. Data can be available upon request to riccardo.dambrosi@hotmail.it.

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
