# Peer review of "Octogenarians Are the New Sexagenarians: Cruciate-Retaining Total Knee Arthroplasty Is Not Inferior to Posterior-Stabilized Arthroplasty in Octogenarian Patients"

_jcm, 2022, doi:10.3390/jcm11133795_

Round 1

Reviewer 1 Report

During this proposed investigation, the authors tried to compare survivorship and functional results in individuals aged 80 and above who have had two different techniques (CR and PS) in their TKA surgery. Their results demonstrate similar clinical outcomes in both approaches with little favor to the CR technique, mainly based on surgical time.  I suggest the following:
1) Schematic graf explaining the surgical technique is helpful for some readers out of the field.
2)  A bit of statistics on people who undergo TKA after the 80s. 

Author Response

1) Schematic graf explaining the surgical technique is helpful for some readers out of the field.

A brief description of the technique has been added as request

Each patients underwent plain radiographies and magnetic resonance imaging (MRI) pre-operatively. Patients were allocated to receive PS or CR. The allocation followed the anatomopathological features of PCL at imaging, clinical, and intra-operative findings. If PCL was intact and effective a CR was implanted, in case of lack or deficient PCL a PS was implanted. A standard medial para-patellar approach was used in all patients. No tourniquet was used. All patients received a cemented Vanguard Knee System (Zimmer-Biomet, Warsaw, IN, USA) following manufacturer instructions. Both tibial and femoral components were cemented using Refobacin® Bone Cement R (Zimmer Biomet, Warsaw, Indiana, USA) [3]."

2) A bit of statistics on people who undergo TKA after the 80s. 

This has been added in introduction section

" According to Swedish Knee Arthroplasty Register, the incidence of knee arthroplasty in patients aged 75 to 84 years old was 665.2/100’000 inhabitatns for women and 546.4/100’000 for men [31]."

Reviewer 2 Report

The primary goal of this study was to compare survivorship and functional results in individuals aged 80 and above who have had total knee arthroplasty (TKA) with cruciate-retaining (CR) or posterior-stabilized (PS) implants. Methods: We prospectively analyzed clinical records of two consecutive cohorts for a total of 96 implants of patients aged 80 years or above. The first cohort consisted of 59 consecutive cemented PS, while the second cohort comprised 37 consecutive cemented CR. The decision to either perform a PS or CR arthroplasty was taken based on pre-operative magnetic resonance imaging and intra-operative findings. The clinical evaluation entailed evaluating each patient’s visual analog scale for pain (VAS), range of motion (flexion and extension), Knee Society Score (KSS), and Oxford Knee Score (OKS). Each patient was clinically evaluated on the day before surgery (T0) and at two consecutive follow-ups at least 1 (T1) and 2 (T2) years after surgery. Implant survival was calculated using the Kaplan–Meier method. Results: Both groups showed a statistically significant improvement at each follow-up compared with the pre-operative value (p <0.05). CR showed a higher flexion degree at T1 compared with the PS group (116.14° ± 5.57° versus 113.16° ± 7.66°; p = 0.048). No differences were found between the two groups regarding survival rate (Chi-squared test p-value = 0.789). Three failures were noted in the CR group, while there were four in the PS group. Conclusions: This prospective clinical study demonstrates that CR and PS TKA have similar clinical outcomes in octagenarians concerning knee function, postoperative knee pain, and other complications. Prosthesis survivorship for CR and PS TKA are both satisfactory, and in selected octagenarian patients, CR should always be considered because of the reduced surgical time.

In General: it's a good paper and the subject of the manuscript is applicable and useful. 

Title: the title properly explains the purpose and objective of the article

Abstract: abstract contains an appropriate summary for the article, the language used in the abstract is easy to read and understand, and there are no suggestions for improvement.

Introduction: authors do provide adequate background on the topic and reason for this article and describe what the authors hoped to achieve.

Results: the results are presented clearly, the authors provide accurate research results, and there is sufficient evidence for each result.

Conclusion: in general: Good research provides sample data for the authors to conclude.

Grammar: Need Some revision.    

Finally, this was an appealing article, in its current state it adds much new insightful information to the field. 

Author Response

In General: it's a good paper and the subject of the manuscript is applicable and useful. 

Title: the title properly explains the purpose and objective of the article

Abstract: abstract contains an appropriate summary for the article, the language used in the abstract is easy to read and understand, and there are no suggestions for improvement.

Introduction: authors do provide adequate background on the topic and reason for this article and describe what the authors hoped to achieve.

Results: the results are presented clearly, the authors provide accurate research results, and there is sufficient evidence for each result.

Conclusion: in general: Good research provides sample data for the authors to conclude.

Grammar: Need Some revision.    

Finally, this was an appealing article, in its current state it adds much new insightful information to the field. 

Thanks for your revision.We correct grammatical error as requested
